# Enhanced Automatic Span Segmentation of Airborne LiDAR Powerline Point Clouds: Mitigating Adjacent Powerline Interference

**DOI:** 10.3390/s25206448

**Published:** 2025-10-18

**Authors:** Yi Ma, Guofang Wang, Tianle Liu, Yifan Wang, Hao Geng, Wanshou Jiang

**Affiliations:** 1Electric Power Research Institute, Yunnan Power Grid Company Ltd., Kunming 650217, China; mayi@dlyjy.yn.csg.cn (Y.M.); wangguofang@dlyjy.yn.csg.cn (G.W.); wangyifan@yn.csg.cn (Y.W.); genghao@dlyjy.yn.csg.cn (H.G.); 2State Key Laboratory of Information Engineering in Surveying Mapping and Remote Sensing, Wuhan University, Wuhan 430079, China; tianleliu@whu.edu.cn

**Keywords:** airborne LiDAR point cloud, span segmentation of powerline point cloud, point counting grid, density clustering

## Abstract

Extracting powerline point clouds from airborne LiDAR data and conducting 3D reconstruction has become a critical technical support for automatic transmission corridor inspection. To enhance data processing efficiency, this paper proposes an automatic method for span segmentation of powerline point clouds that accounts for adjacent powerline interference, aiming to provide “clean” data for the automatic reconstruction of powerline catenary curve models of each span. This method tackles a key challenge in airborne LiDAR data: interference from adjacent or cross-over powerlines when automatically extracting main-line pylon positions and powerline points. Leveraging the spatial relationship between pylons and powerlines in LiDAR point clouds, we developed a fast density clustering algorithm based on a novel point-counting grid (PCGrid), which greatly accelerates DBSCAN clustering while adaptively extracting main-line pylons and powerline point clouds. The method proceeds in three steps: first, using 2D density clustering to extract reliable pylon positions and 3D density clustering to filter out non-main-line point clouds; second, verifying pylon connection combinations via main-line point clouds and identifying the longest line in the connection matrix as the pylons of the main powerline; and third, assigning powerline points to their corresponding spans for segmented reconstruction. Experimental results demonstrate that the proposed PCGrid structure not only significantly improves clustering efficiency, but also enables a fully automated span segmentation process that effectively suppresses adjacent powerline interference, highlighting the novelty of integrating efficient PCGrid-based clustering with spatial-relationship-driven pylon verification into a unified framework for reliable 3D powerline reconstruction.

## 1. Introduction

In the present era, China’s economy is experiencing remarkable and rapid growth, with the prosperity of various industries driving a continuous increase in electricity demand. As the core channels for power transmission, powerline corridors have become increasingly complex in their spatial distribution, playing a pivotal role in ensuring the reliable delivery of energy. However, these corridors often traverse mountainous forests and other geographically challenging areas, which has further highlighted the limitations of traditional power inspection methods. Such approaches require inspection personnel to venture into hazardous environments, facing risks posed by complex terrain and potential encounters with wild animals. Moreover, the low efficiency of manual inspections fails to keep pace with the accelerating expansion of electricity demand and the growing scale of power transmission networks [1,2,3].

The advent of airborne Light Detection and Ranging (LiDAR) technology has introduced a transformative force in the field of powerline inspection. Its unique ability to penetrate vegetation canopies enables rapid acquisition of high-precision and high-density three-dimensional spatial information of terrestrial targets, a feature that has led to its widespread application in the surveying and mapping industry. Through the post-processing of point cloud data and three-dimensional reconstruction, powerline models within powerline corridors can be extracted efficiently and accurately. This capability significantly enhances the efficiency of powerline inspections and serves as a critical technological advancement supporting the modernization of the power industry [4,5,6].

Consequently, numerous researchers have focused on developing algorithms for powerline extraction. Guo et al. [7] proposed a supervised classification method using a JointBoost classifier with geometric and echo features, enhanced by feature selection and graph-cut optimization, to accurately classify airborne LiDAR data including powerlines and pylons. Zhou et al. [8] addresses complex scenes with substantial noise by employing a JointBoost classifier that integrates slope, density, and elevation features to separate powerline point clouds, followed by dual RANSAC and catenary models to extract split subconductors. Liang et al. [9] developed a method to automatically extract and reconstruct multiple powerlines from airborne LiDAR data using density clustering and least squares fitting, achieving accurate recovery of their geometric features. Jwa et al. [10] proposed an automated approach that extracts candidate powerline points, fits them to a catenary curve, and progressively refines the model using a stochastic constrained non-linear adjustment. Ortega et al. [11] proposed a seven-stage pipeline for classifying pylons and wires in LiDAR data and modeling powerlines. The method separates different wire and pylon types, identifies insulators, and fits each conductor with a 3D catenary model using particle swarm optimization. Munir et al. [12] proposed an automated method to extract individual subconductors from complex multi-bundle powerline corridors. By combining image- and point-based processing with voxel grouping, statistical analysis, and recursive clustering, the method effectively distinguishes pylons, trees, and conductors, achieving highly accurate extraction of bundle structures in real-world test sites. Yan et al. [13] applies statistical filtering and cloth simulation filtering to preprocess point clouds, then utilizes principal component analysis and fast Euclidean clustering to extract individual powerlines and perform multidimensional fitting. Methods [14,15] use an improved random forest classifier to segment powerline corridor point clouds and apply enhanced density-based clustering to extract powerline points. Yermo et al. [16] introduced a method for detecting and modeling powerlines of any voltage, without requiring targeted flight missions, which includes an improved Hough transform-based detection with robust clustering for conductors sharing the same vertical plane. Cao et al. [17] introduced an automated method for extracting powerlines from ALS point clouds, combining grid-based pylon detection, span-based data organization, and robust segmentation via 3D Alpha Wrapping. Additionally, Methods [18,19,20,21] demonstrate the feasibility and effectiveness of deep learning methods for pylon point cloud extraction.

Although powerline extraction algorithms have made notable progress, practical applications still face critical bottlenecks. The spatial configuration of powerlines follows the physical characteristics of a catenary curve, and constructing its mathematical model strictly depends on the spatial relationship between two adjacent pylons. Therefore, each segment of the catenary model must be accurately derived from its corresponding pair of pylons. However, most studies on powerline modeling implicitly assume that the point cloud data have already been segmented by span [7,13,22,23,24]. If powerlines from different spans are not effectively separated, model fitting becomes disordered, making it impossible to represent the true spatial state of the powerlines. Existing span segmentation methods, however, exhibit significant limitations: some rely on manually designed pylon extraction rules, which are time-consuming, labor-intensive, and procedurally complex [25]; others depend on prior knowledge [1,26,27] or on manual selection of clustering hyperparameters [17], which restricts their applicability in real-world scenarios. Moreover, these methods generally fail to handle interference from pylons belonging to adjacent powerlines [10], thereby limiting the level of automation achievable in three-dimensional powerline modeling.

In this study, we address the core problem of automatic span segmentation of powerline point clouds, particularly when precise pylon coordinates are unavailable. The proposed method first introduces the Point Count Grid (PCGrid) to support efficient density-based clustering, enabling the extraction of pylon centers and main-line powerline points from complex point clouds that may contain intersecting or parallel powerlines as well as false pylon candidates. Next, the extracted powerline points are used to verify pylons and their connectivity, ensuring that only valid pylons along the main line are retained. Finally, each powerline point is automatically assigned to the span defined by its nearest pair of connected pylons, thereby producing clean span-level segmentation suitable for subsequent catenary model construction. Experiments on three real-world datasets with powerline interference validate the effectiveness of the proposed approach. The main innovations of this work are as follows:
The introduction of the Point Count Grid (PCGrid) to significantly accelerate density-based clustering of large-scale point clouds.A pylon verification and connection validation strategy that robustly distinguishes main-line pylons from interference pylons.A fully automated span segmentation framework that directly outputs span-level powerline point clouds, laying a reliable foundation for accurate 3D catenary reconstruction.

## 2. Methods

### 2.1. Automatic Span Segmentation of Powerlines

#### 2.1.1. Experimental Data and Preprocessing

To evaluate the generalizability of the proposed method, this study employs point cloud datasets of three powerline corridors (35 kV, 110 kV, and 220 kV) located in Yunnan Province as the original data. The datasets have been preprocessed through deep learning-based classification, achieving a classification accuracy exceeding 95% for both powerlines and pylons. Both selected powerlines contain multiple instances of crossing spans and interference from nearby powerlines, making it challenging to directly identify the pylons and powerlines belonging to the main line. The 220 kV powerline comprises a total of 42 pylons, the 110k V powerline comprises 36 pylons, and the 35 kV powerline comprises 28 pylons. All the data were collected using DJI Inspire M350 and Digital Green X4 LiDAR.

To enhance the level of automation and minimize manual intervention, the processing in this study is performed on powerline and pylon point clouds extracted from the results of deep learning-based classification. The workflow does not involve any manual cropping or denoising operations; instead, noise handling is fully integrated into the processes of pylon detection and extraction, pylon verification, and automatic span segmentation of powerlines. The common shape of the powerline and pylon point clouds are listed in Figure 1, and the noise in the dataset is demonstrated in Figure 2. We consider the classified non-main-line powerlines and pylons point clouds as noise.

#### 2.1.2. Overall Processing Workflow

The overall processing workflow of this study is divided into two main components: pylon center extraction and verification, and powerline point cloud span segmentation, with pylon center extraction and verification constituting the core step. Pylon detection is performed on the classified pylon point clouds using the DBSCAN density-based clustering algorithm to extract individual pylon clusters. Pylon verification leverages the inherent relationships between pylons and powerlines, as well as the consistency between pylon connections and powerline geometry. The span segmentation of powerline point clouds is then conducted based on the correctly and sequentially connected pylons. By calculating distances between powerline points and the connecting lines between pylons, powerline points are assigned to their corresponding spans. The overall processing workflow is illustrated in Figure 3.

To accelerate the DBSCAN density clustering, this study proposes two- and three-dimensional Point Count Grids, PCGrid_2D and PCGrid_3D, upon which the point cloud clustering and kd-tree structures are constructed. Compared with conventional DBSCAN implementations based on kd-trees, the proposed method achieves a significant improvement in computational efficiency.

### 2.2. Density Clustering of Pylons and Powerlines Based on the Point Count Grid

#### 2.2.1. Point Count Grid (PCGrid)

To address the high computational cost of the DBSCAN algorithm and the persistent time-consuming issues even after kd-tree acceleration in clustering high-density point clouds, this study proposes a Point Count Grid (PCGrid) data structure to efficiently accelerate DBSCAN. As illustrated in Figure 4, the two-dimensional Point Count Grid (PCGrid_2D) divides the data range into a two-dimensional grid according to a specified interval. It then records the number of points falling within each grid cell and calculates the weighted centroid of points in each cell (represented by the red dots in Figure 4) by averaging their positions. This centroid serves as a representative location for the points within the cell. The construction of the three-dimensional Point Count Grid (PCGrid_3D) follows a similar approach to that of the two-dimensional grid.

To improve memory efficiency, a sparse matrix representation is employed to organize the grid, wherein only cells containing points are recorded and stored using std::map, trading computation time for reduced memory usage. For a two-dimensional grid with $N_{y}$ rows and $N_{x}$ columns, the index ID of the grid cell at coordinates $\left(i_{x},i_{y}\right)$ is given by(1)$$\mathrm{ID}=i_{x}+i_{y}\times N_{x}$$

Similarly, for a three-dimensional grid with $N_{z}$ layers, $N_{y}$ rows, and $N_{x}$ columns, the index ID of the grid cell at coordinates $\left(i_{x},i_{y},i_{z}\right)$ is given by(2)$$\mathrm{ID}=i_{x}+i_{y}\times N_{x}+i_{z}\times N_{y}\times N_{x}$$

The neighborhood system of PCGrid_2D is defined as the 8-neighborhood (3×3−1), while that of PCGrid_3D is the 26-neighborhood (3×3×3−1). Neighboring grid cells are queried based on their IDs stored in the std::map.

#### 2.2.2. Pylon Point Cloud Clustering and Center Extraction Based on 2D PCGrid DBSCAN

A Point Count Grid (PCGrid_2D) is constructed with a 3-m interval, onto which all pylon point clouds are subsequently registered. This allows for counting the number of points within each grid cell, which forms the basis for performing DBSCAN clustering on the pylon point clouds. The specific procedure is as follows:All non-empty grid cells containing point clouds are sorted based on their point counts;DBSCAN clustering is performed in descending order of the number of points within each grid cell;Unlike the standard DBSCAN algorithm, the minimum number of points criterion is applied solely based on the points contained within each individual grid cell in the two-dimensional PCGrid_2D.

For the pylon point clouds obtained through clustering, the centroid position as well as the maximum and minimum elevations are calculated. Additionally, the covariance matrix is employed to determine the principal orientation and the semi-axes lengths of the pylons, providing essential data for subsequent processing.

#### 2.2.3. Extraction of Main-Line Powerline Point Clouds Based on 3D Point Count Grid DBSCAN

A three-dimensional Point Count Grid (PCGrid_3D) is constructed with a 3-m interval. Subsequently, all powerline point clouds are registered onto this 3D grid to obtain the number of points within each grid cell. Based on this, DBSCAN clustering is performed on the powerline point clouds to extract the main-line powerline point clouds.

Unlike pylon point cloud clustering, due to the sparsity of powerline point clouds, the minimum number of points criterion in the three-dimensional PCGrid_3D must account for the points in neighboring grid cells. For continuous powerline point clouds, only the cluster with the highest point count needs to be considered when extracting the main-line powerline point clouds.

### 2.3. Extraction of Main-Line Pylons Based on the Relationship Between Pylons and Powerlines

#### 2.3.1. Initial Pylon Ordering Based on Nearest Distance

To simplify the subsequent tracking of the main line in the connection matrix, pylons are first ordered based on their spatial proximity, forming a linked list of pylons starting from the designated departure point. If the departure point is known, it can be set as the virtual starting point, either the known pylon or the first point in the powerline point cloud, and the nearest pylon is iteratively identified to construct an ordered list of pylons beginning from this point. In cases where the pylon point cloud is merged from multiple flights, the first point of the point cloud may not correspond to the initial pylon. In such situations, the starting point can be selected from one of the two pylons with the greatest pairing distance.

During the formation of the ordered list, exceptions may arise where the first pylon is not actually the true initial pylon. In such cases, the preceding pylons may be linked after the last pylon in the sequence. To address this, anomaly detection is necessary to reconnect the pylons separated by the unusually long distance back before the first pylon. The algorithm proceeds as follows:Select a virtual starting point and identify the pylon closest to this point as the first pylon;From the remaining pylons, find the pylon nearest to the current pylon;Swap the nearest pylon to the position immediately following the current pylon, then update this pylon as the current pylon;Repeat step (2) until all the pylons have been processed;Starting from the last pylon in the ordered list, compare each pylon sequentially with the first pylon;If the distance between a pylon and the first pylon is less than the distance between this pylon and its preceding pylon, relocate this pylon and all subsequent pylons to precede the first pylon;Output the reordered list of pylons.

As illustrated in Figure 5, consider pylons a, b, c, …, n. If the first pylon is initially assigned to pylon d, the nearest-neighbor sorting yields the sequence defghijklmcba. Starting from the last pylon in the sorted list (pylon a), the distance between each pylon and the first pylon d is checked sequentially. It is observed that the distance between pylon c and pylon d is less than the distance between pylon d and pylon m. Consequently, pylons c, b, a are reversed and placed before pylon d, resulting in the correctly ordered pylon sequence: abcdefghijklm.

#### 2.3.2. Pylon Filtering Based on the Relationship Between Pylons and Surrounding Powerlines

Pylons outside the main-line powerline can be effectively filtered out by leveraging the spatial relationship between pylons and powerlines. As illustrated in Figure 6, from the perspective of spatial distribution, pylons within the main line are typically enclosed by powerlines in the horizontal plane, and their tops are higher in elevation than the surrounding powerlines. Based on this observation, for each pylon, a neighborhood search is performed to retrieve nearby powerline point clouds. Subsequently, an analysis of the horizontal and elevation relationships is conducted to determine whether the pylon belongs to a non-main line. The detailed algorithm is as follows:

Construct a horizontal grid PCGrid_2D with a 1-m interval based on the spatial extent of the powerline point cloud, and register the powerline points into this grid;Retrieve the centroid coordinates of grids containing powerline points and insert them into a kd-tree to build a two-dimensional spatial index;For each pylon, use its center point as the reference and query the kd-tree within a specified radius to obtain neighboring grid points and their corresponding original LiDAR points;Perform principal component analysis (PCA) on the neighboring LiDAR points to extract the centroid, eigenvalues, eigenvectors, and the lengths of the major and minor axes;Compute the projection of the pylon center onto the eigenvectors;If the distance between the pylon center and the centroid of the powerline point cloud exceeds the length of the minor axis, the pylon is classified as either an outlier pylon or noise outside the main-line powerline.

In this study, the neighborhood radius is set to 6 m.

#### 2.3.3. Validity Assessment of Pylon Connections Based on Powerline Point Clouds

Apart from the main-line powerline, the LiDAR-scanned point clouds often contain pylons from adjacent or intersecting powerlines. As a result, the distance-based sorted pylon list may include non-main-line pylons, leading to incorrect connections. For example, pylon k in Figure 5 belongs to an interfering powerline.

To eliminate pylons from interfering powerlines, we propose a pylon connection validation method based on powerline point clouds. As illustrated in Figure 7, the conductors between pylons $T_{i}$ and $T_{j}$, with radii $r_{i}$ and $r_{j}$, respectively, are located within an approximate trapezoidal region. If $T_{i}$ and $T_{j}$ are two consecutive pylons on the same powerline, powerline point clouds are expected to be evenly distributed along the connection between them. Based on this spatial relationship between pylons and powerlines, we develop an algorithm to verify whether a pylon connection is valid.

Construct a 2D point counting grid (PCGrid_2D) with a grid interval of 3 m;Extract the grid centroids and build a kd-tree for these centroid points;Sample points along the powerline segment $T_{i},\;T_{j}$ at the grid interval, and interpolate the radius $r_{ij-k}$;For each sampled point, search for the nearest point in the kd-tree;If the distance $d_{ij-k}$ from the sampled point to the nearest grid centroid is less than $r_{ij-k}$, and the powerline point cloud at this position is not higher than the connecting line of adjacent pylons, mark this sampled point as valid;Repeat steps (3) to (5) until all sampled points are processed;If the proportion of valid sampled points exceeds 90%, the pylon connection $T_{i},\;T_{j}$ is deemed valid and likely represents a real pylon linkage.

This algorithm can determine whether there is sufficient powerline presence between two pylons, indicating a possible pylon connection. However, there is an exception: for pylons aligned in a straight powerline, any pair of pylons may satisfy this condition. Therefore, it is necessary to further verify connections based on the pylon connection matrix combined with the pylon ordering.

#### 2.3.4. Extraction of the Main-Line Powerline Based on the Pylon Connection Matrix

Since the pylons have already been sorted based on distance, powerline tracing can be performed by following the nearest neighbor principle among pylons connected by powerlines. As illustrated in Figure 8, pylon 1 is connected to pylons 4 and 6, and by applying the nearest neighbor rule, the connection 1–4 is established. Among the four pylons, only pylon 6 is connected, resulting in the sequence 1–4–6. Pylon 6 is connected to both pylons 7 and 8; selecting pylon 7 yields the sequence 1–4–6–7. Pylon 7 is connected only to pylon 8, extending the sequence to 1–4–6–7–8. Finally, pylon 8 connects solely to pylon 9, completing the sequence 1–4–6–7–8–9. Therefore, the longest valid connection within this group of nine pylons includes six pylons, with pylons 2, 3, and 5 identified as interference pylons.

### 2.4. Fine Segmentation of Powerline Point Clouds

Point cloud segmentation involves assigning each LiDAR point of the main-line powerline to the segment between the two corresponding pylons it belongs to. First, a kd-tree is constructed to index the pylon centers. Then, for each laser point, the nearest pylon is queried, and it is determined whether the point lies to the left or right of that pylon. The detailed steps are as follows:Construct a 2D kd-tree index for the pylon centers;For each powerline point, query the nearest pylon *i* using the 2D kd-tree;If the point lies between the line segment connecting pylons i−1 and *i*, assign the point to the segment (i−1,i);Otherwise, if the point lies between the line segment connecting pylons *i* and i+1, assign the point to the segment (i,i+1);Repeat steps (2) to (4) until all points have been processed.

To determine whether a powerline LiDAR point lies between pylons *i* and i+1, the approach illustrated in Figure 7 is employed. Specifically, the (X,Y) coordinates of the point are transformed into the coordinate system defined by the line segment connecting pylons *i* and i+1. Then, the point’s position is evaluated based on its longitudinal offset (distance along the line) and lateral distance (perpendicular offset) to decide if it falls within the segment boundaries.

## 3. Experiments and Results

### 3.1. 220 kV Powerline

The powerline point cloud dataset comprised 4,966,307 points, as shown in Figure 9, which were mapped to 75,790 grid cells and subsequently clustered into 74 distinct powerline groups, of which 4,744,580 points belonged to the main-line powerline. The pylon point cloud dataset contained 1,922,359 points, distributed across 139 grid cells. For the initial pylon extraction, density-based clustering identified 61 candidate pylons, from which structures with a height of less than 3 m were excluded, yielding 58 pylons. Pylon verification, based on both horizontal and elevation relationships, identified 42 pylons located along the powerline, while removing nine interference pylons. Using the pylon connection matrix, the main-line powerline was determined to comprise exactly 42 pylons, consistent with the actual number of pylons present in the surveyed area. The final span segmentation result is shown in Figure 10.

Among the nine excluded interference pylons, two primary types of interference were identified: interference from pylons belonging to adjacent powerlines and interference from pylons located on intersecting lines. The following section provides a detailed analysis of representative interference cases for each of these two categories.

#### 3.1.1. Interference from Adjacent Lines

After point cloud classification, some non-main-line powerlines will appear near the main-line powerlines. This phenomenon usually occurs at the branching points of the powerlines. Our method is capable of handling the interference from these parallelly distributed adjacent lines. Figure 11 demonstrates that our method is robust to extract the main line with the interference of adjacent lines. In Figure 11a, although there are three pylons for detecting other lines beside the main-line powerline, during the subsequent connectivity test, these power towers can be filtered out. In Figure 11b, although there is one pylon on the left that is mistakenly detected as a powerline, this noise can be filtered out in the subsequent powerline clustering process.

#### 3.1.2. Interference from Crossing Lines

Figure 12 demonstrates that our method can accurately extract the main line without being disturbed by the intersecting lines. In Figure 12, there will be a large number of intersecting powerlines and pylons beside the main-line powerline. And these point clouds, like the main-line point clouds, are also classified into the same category. These point clouds will significantly interfere with the correct extraction of the main line. Our method can robustly achieve the correct segmentation of the main line.

### 3.2. 110 kV Powerline

The powerline point cloud dataset contains a total of 9,460,980 points, as shown in Figure 13, which were registered into 42,392 grids and clustered into 79 distinct powerline groups, among which 7,886,718 points belong to the main-line powerline. The pylon point cloud dataset comprises 8,861,405 points, registered into 345 grids. Density-based clustering identified 221 pylons, and after removing pylons with heights below 3 m, 90 valid pylons were retained. Pylon verification, based on horizontal and elevation relationships, identified 40 pylons within the powerline, while 50 interference pylons were removed. Using the connection matrix, 36 main-line pylons were obtained, with an additional 14 interference pylons eliminated, yielding a count consistent with the actual number of pylons. The span segmentation result is shown in Figure 14.

#### Local Analysis

Figure 15 shows that our method can also be applied in powerline corridors with lower voltages, demonstrating the universality of the method. The 110 kV voltage falls under the lowest voltage category in high-voltage power transmission. The power pylons of 110 kV powerlines are relatively short, and the influence of ground features on classification accuracy is significant. There is an increase in interference lines and noise, which poses higher requirements for accurately extracting the main lines. Our method can accurately obtain the main line and span segmentation information in an extremely short time.

### 3.3. 35 kV Powerline

The 35 kV transmission voltage no longer falls under the category of high-voltage transmission. In the main lines of this voltage level, there will be many pylons with a height of approximately 10 m even less. These power towers are highly coupled with the terrain features, and after point cloud classification, there is a large amount of noise. In the 35 kV data, the powerline point cloud contains 1,719,082 points, which are mapped into 15,794 grid cells, and subsequently clustered into 349 groups. In total, 1,283,720 points belong to main-line powerline point cloud. The pylon point cloud contains 4,293,888 points, which are mapped into 439 grid cells and initially clustered into 164 pylon centers. As shown in Figure 16 after verification, the classification algorithm outputs 28 main-line pylons and the span segmentation identifies 27 pylons with correct main-line span. Due to the low height of one of the pylons, the point cloud was incomplete. The span segmentation identified this pylon as the powerline and connected the two adjacent spans, resulting span merging.

In Figure 17, due to the significant terrain height variations in these data, the classification algorithm mistakenly identified ground or vegetation point clouds as pylons or powerlines, which posed difficulties in the span segmentation process. The span segmentation algorithm will generate a large number of false powerlines and false pylons, but these noises will be filtered out during the verification based on the relationship between the pylons and powerlines.

### 3.4. Efficiency Analysis

The segmentation time was averaged over three runs, excluding the time required to read the point cloud data. The total segmentation time for the 220 kV powerline was 7.3 s, while for the 110 kV line, it was 8.8 s.

Similarly, as shown in Table 1, the pylon clustering time was averaged over three runs, excluding point cloud loading time. As shown in Table 2, for the 220 kV line with 1,922,359 pylon points, the clustering time accelerated by a kd-tree algorithm was 287.3 s, whereas the PCGrid-accelerated clustering took only 0.42 s, resulting in an approximate speedup factor of 684. For the 110 kV line with 8,861,405 pylon points, the kd-tree-accelerated clustering required 2251.6 s, while PCGrid acceleration reduced this to 2.57 s, achieving a speedup of about 876 times.

From an implementation perspective, traditional DBSCAN clustering typically applies point cloud downsampling before clustering to reduce computation time; however, the downsampling ratio heavily depends on prior knowledge of the data and involves considerable manual tuning. The point counting grid proposed in this work can be regarded as an adaptive downsampling method that effectively achieves a high downsampling rate while retaining comprehensive point information.

### 3.5. Robustness Analysis

In this section, we discuss the performance of our method by adding more noise in the original dataset. Since the original dataset is derived from point cloud classification predictions, there is some inherent noise. However, the structures of the main-line powerlines and pylons are relatively clear. We tested the robustness of our method by randomly deleting points from the point cloud.

As the result in Table 2, although we randomly delete 5% and 10% of the total point clouds of 220 kV and 110 kV datasets, the number of main-line pylons detected is consistent with the original dataset and accurate.

## 4. Conclusions

This paper presents a fully automated method for extracting main-line pylons and segmenting powerlines from LiDAR point clouds classified via deep learning. The method effectively handles interference from adjacent lines without requiring prior knowledge of pylon coordinates. By combining density-based clustering with spatial-relationship analysis, it accurately identifies main-line pylons, suppresses noise and interference, and assigns powerline points to their corresponding spans. Furthermore, the proposed Point Count Grid (PCGrid) greatly accelerates clustering, enhancing both efficiency and practical applicability for 3D powerline reconstruction. The algorithm we proposed can be easily integrated into software and used as a post-processing step for power corridor data, possessing potential engineering application value.

## 5. Future Work

Future work will focus on enhancing the robustness and generalizability of the proposed method through multi-scale PCGrid clustering and confidence-based error correction, while integrating deep learning approaches with rule-driven strategies to jointly leverage semantic features and spatial constraints for more reliable span segmentation and pylon extraction.

## Figures and Tables

**Figure 1 sensors-25-06448-f001:**
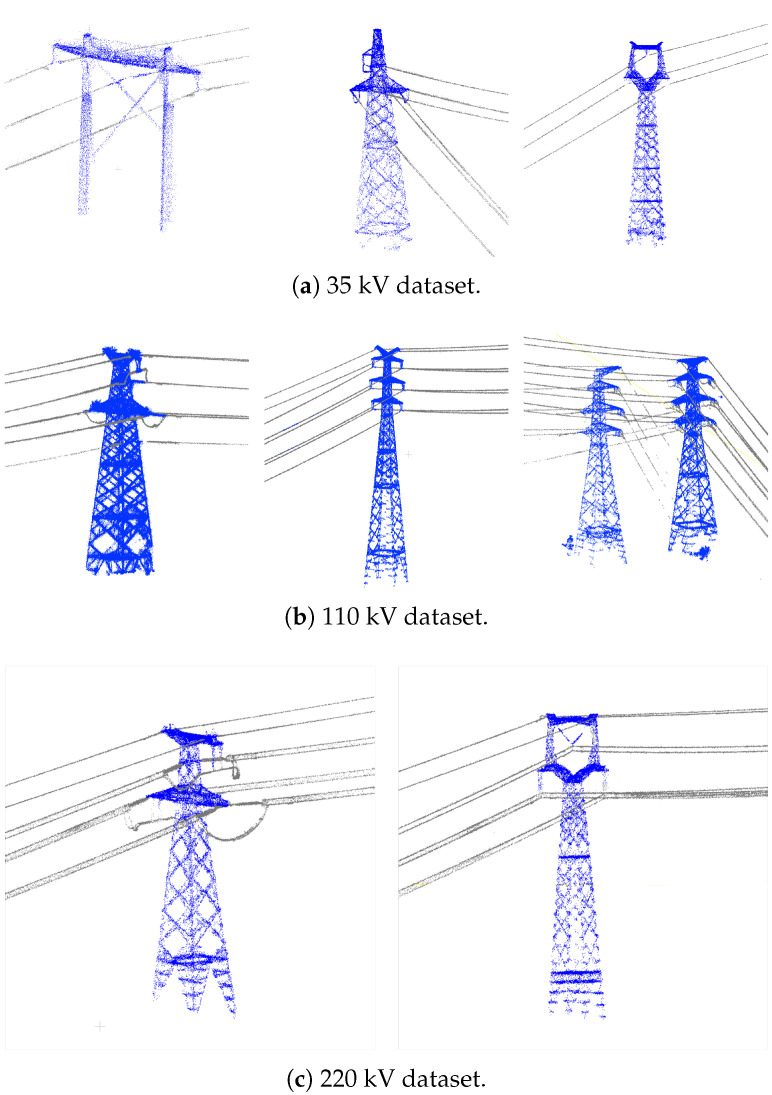
Common shapes of pylons and lines in the test dataset. The blue points represent pylon point clouds, and the gray points represent powerline point clouds.

**Figure 2 sensors-25-06448-f002:**
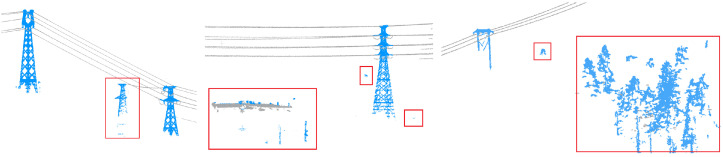
Original noise in the dataset. The points in the red box are prediction noise from classification or non-main-line powerlines or pylons, which needs to be removed during the process.

**Figure 3 sensors-25-06448-f003:**
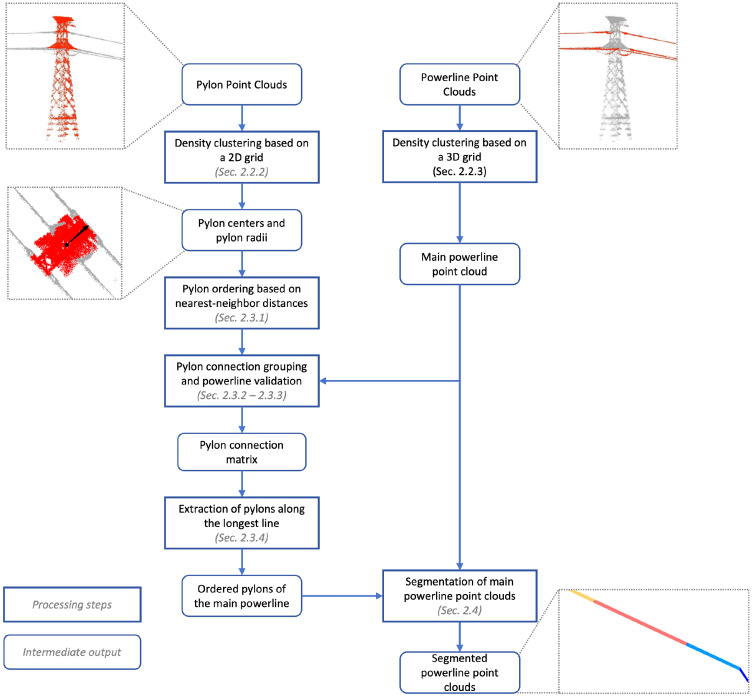
Pipeline of the proposed method. The rounded rectangle frame represents the intermediate output, while the rectangular frame represents the processing step. Our method takes the classified powerline point cloud and pylon point cloud as the input, and outputs the main-line powerline point cloud with span information and the positions of the center of every main-line pylon.

**Figure 4 sensors-25-06448-f004:**
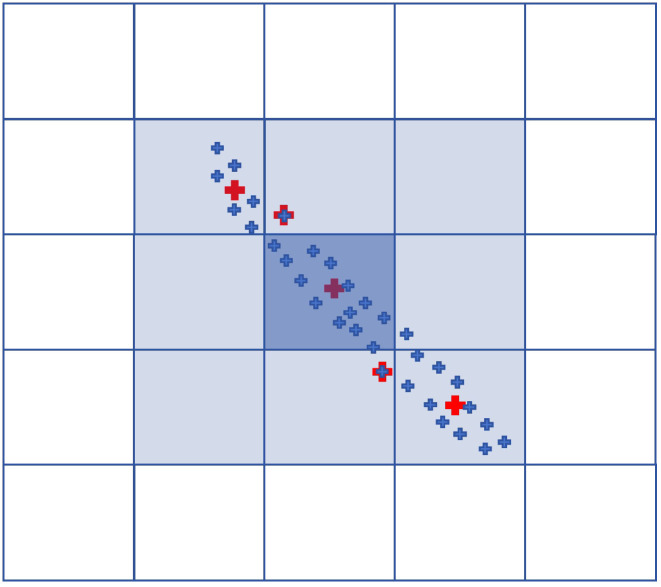
Point counting grid (PCGrid). The small blue crosses represent the point clouds, while the small red crosses represent the weighted centroids within the current grid. When there is exactly one point cloud within the current grid, then the weighted centroid coincides with this point.

**Figure 5 sensors-25-06448-f005:**
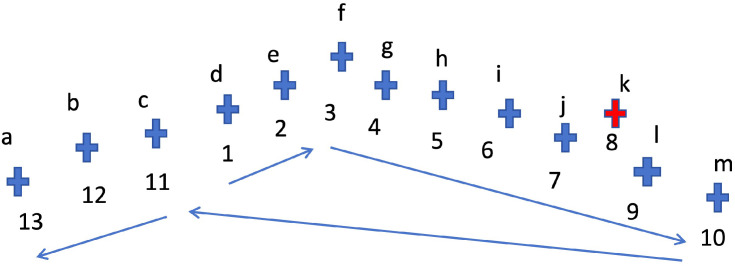
Pylon ordering based on nearest-neighbor distances, in which each cross represents one pylon position. Assuming that cross c is taken as the starting position of the main-line pylon sequence, then the main-line pylon sequence obtained according to the nearest neighbor rule is as shown by the blue arrow and the index number. Among them, the red cross represents one position of the non-main-line pylons.

**Figure 6 sensors-25-06448-f006:**
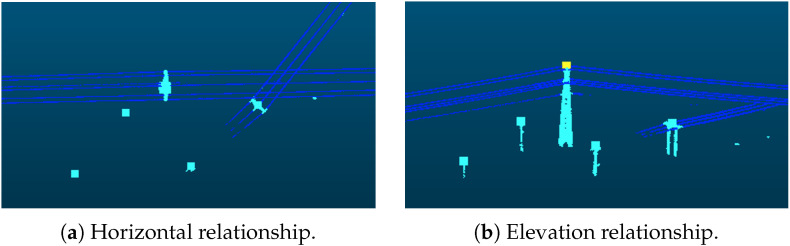
The relationship between pylon and powerline points. The point clouds of the main-line powerlines and pylons will have connectivity. Our method detects the distance between the pylon positions and the powerlines in both (**a**) horizontal and (**b**) vertical directions to filter out the pylons those have no connection to powerlines in the surrounding area.

**Figure 7 sensors-25-06448-f007:**
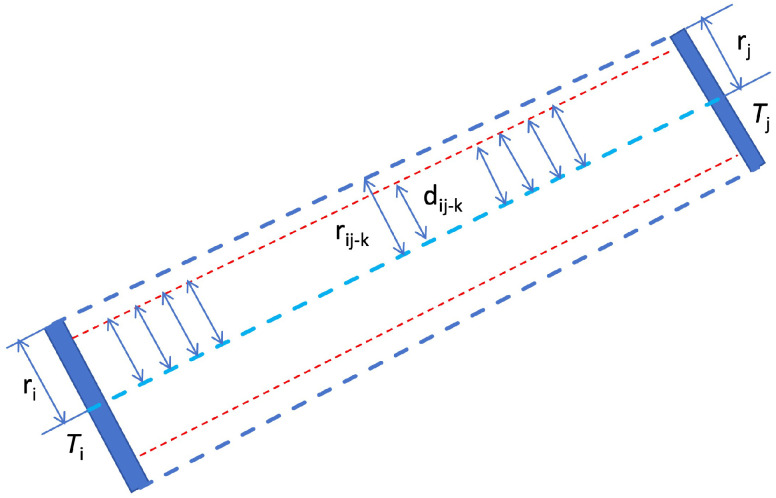
Validity assessment of pylon connections based on powerline point clouds. Our method verifies the connectivity between the two pylons by sampling the points in the channel between them and checking whether there are powerline point cloud points near these points.

**Figure 8 sensors-25-06448-f008:**
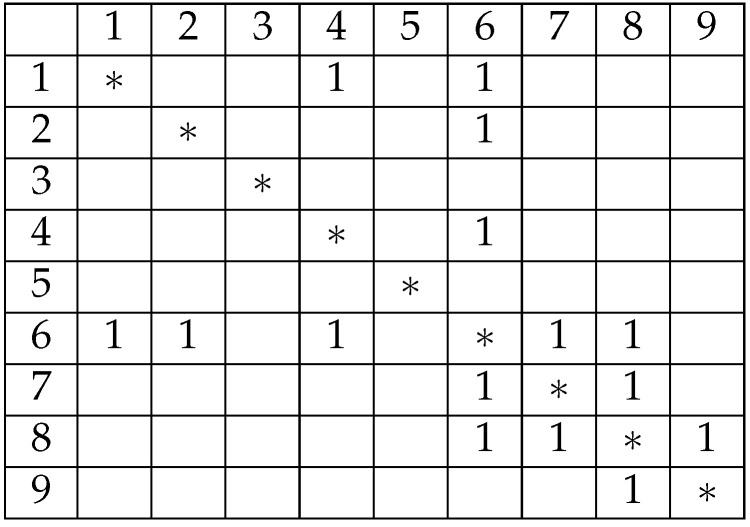
Main-line pylon extraction based on a connection matrix. The numbers represent pylons serial, the 1 in the matrix means is connective.

**Figure 9 sensors-25-06448-f009:**
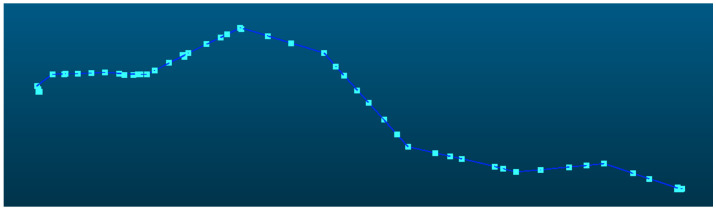
Initial pylons and powerline point clouds. The points and lines represent pylon and span position after initial detection.

**Figure 10 sensors-25-06448-f010:**
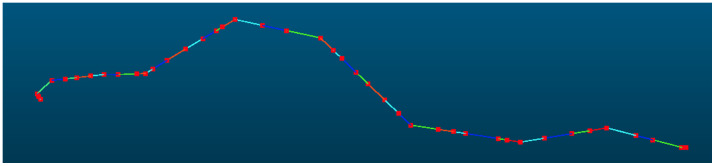
Verified pylons and segmented point clouds. The red points represent verified pylon positions and the colorful lines represent each span.

**Figure 11 sensors-25-06448-f011:**
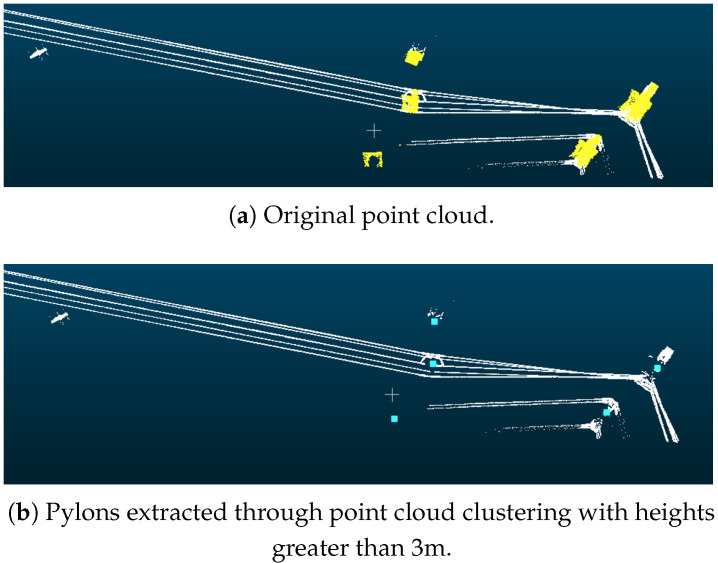
The results of interference from adjacent lines near the start of the main line

**Figure 12 sensors-25-06448-f012:**
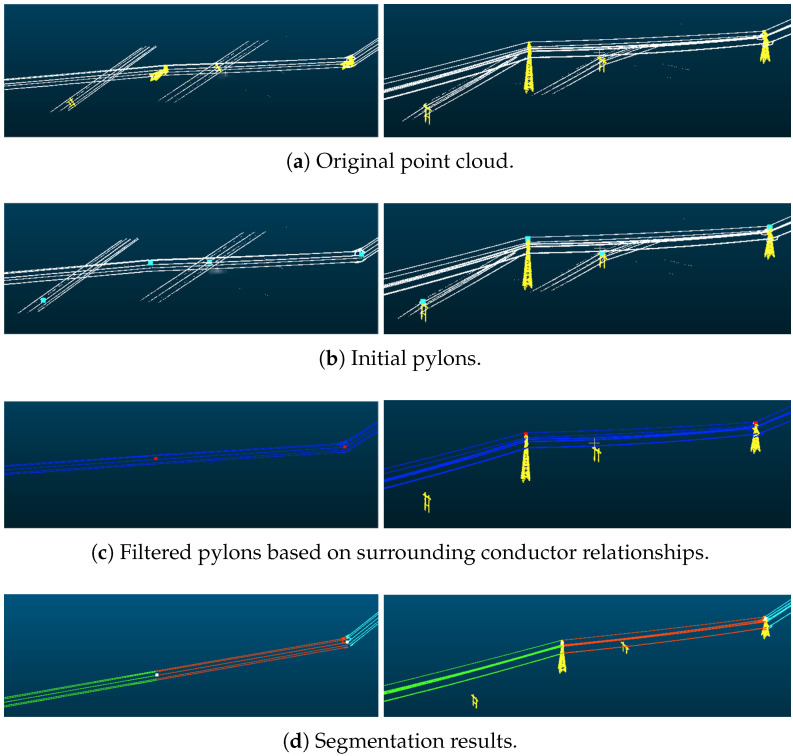
The results of mid-line interference from crossing powerlines.

**Figure 13 sensors-25-06448-f013:**
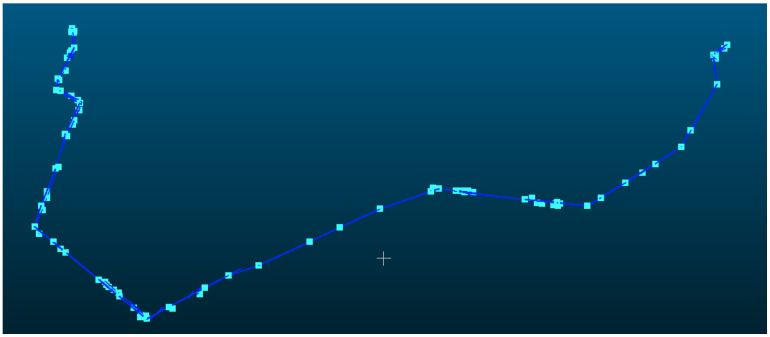
Initial pylons and powerline point clouds. The points and lines represent pylon and span position after initial detection.

**Figure 14 sensors-25-06448-f014:**
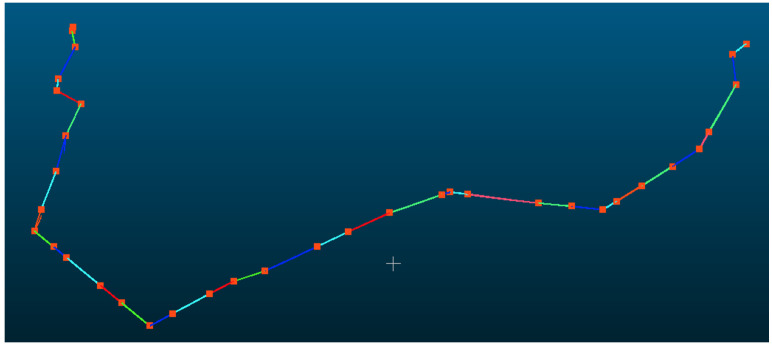
Verified pylons and segmented point clouds. The red points represent verified pylon positions and the colorful lines represent each span.

**Figure 15 sensors-25-06448-f015:**
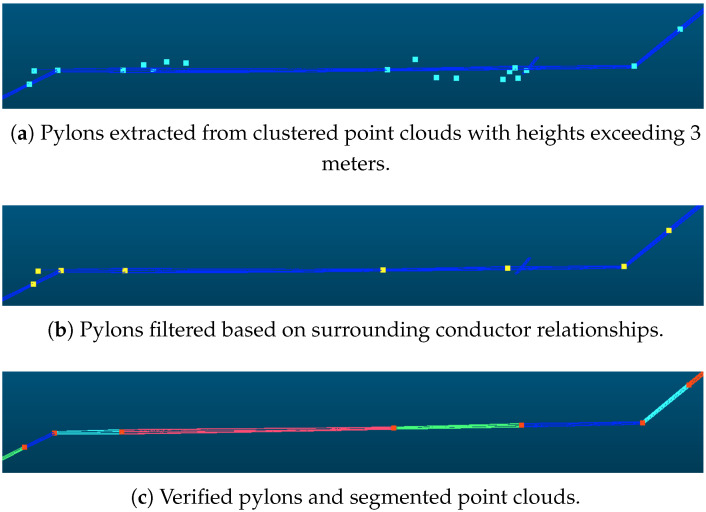
Local results on 110 kV dataset.

**Figure 16 sensors-25-06448-f016:**
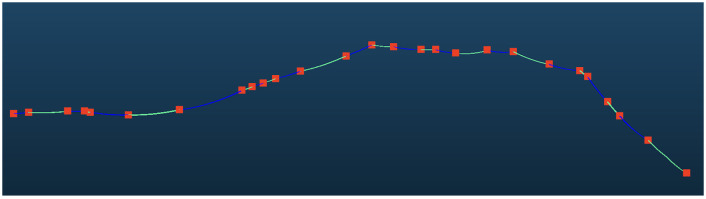
VeriAed pylons and segmented point clouds. The red points represent verified pylon positions and the colorful lines represent each span.

**Figure 17 sensors-25-06448-f017:**
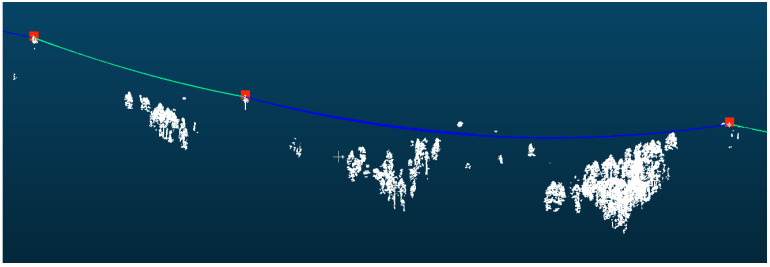
The span segmentation results when the noise level is high. White point clouds include the noise from classification.

**Table 1 sensors-25-06448-t001:** Comparison of pylon clustering time. Our method achieved a speedup ratio of hundreds of times.

	Pylon Points	kd-tree Acceleration (s)	PCGrid Number	PCGrid Acceleration (s)	Acceleration Ratio
220 kV	1,922,359	287.3	139	0.42	**684.0**
110 kV	8,861,405	2251.6	345	2.57	**876.1**

**Table 2 sensors-25-06448-t002:** Comparison of the number of main-line pylons detected.

	Original Data	w/5% Random Deletion	w/10% Random Deletion
220 kV line	42	42	42
110 kV line	36	36	36

## Data Availability

The datasets presented in this article are not readily available because the data are part of an ongoing study. Requests to access the datasets should be directed to wangguofang@dlyjy.yn.csg.cn.

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
