# Peer review of "Enhanced Automatic Span Segmentation of Airborne LiDAR Powerline Point Clouds: Mitigating Adjacent Powerline Interference"

_sensors, 2025, doi:10.3390/s25206448_

Round 1
Reviewer 1 Report
Comments and Suggestions for Authors
Dear authors,
I found the article interesting, and I understand that your work is focused on automatic 3D reconstruction of power lines.
I do not see any novelty in your work, and I believe that the level of innovation is limited. I appreciate your work, and I have the following appreciation and recommendations for improvement.
The abstract is ok; you present your work using enough details, and you present the main scope. I suggest emphasizing what is new.
Regarding the introduction, it is excellent that the reader can understand the importance of power transmission. The four phrases require references to support the opinions you have introduced.
At the final part of the chapter, you present the problem that you solve and some ideas regarding the methodology.
The chapter “Methods” is clear, and you have presented enough details regarding your work.
There is a clear presentation of the experimental results.
In conclusion, you can present some ideas regarding future work.
Author Response
Comment 1: The abstract is ok; you present your work using enough details, and you present the main scope. I suggest emphasizing what is new.
Response 1: Thank you for the comment. We have revised the abstract to make the key novelty point more prominent.
Comment 2: Regarding the introduction, it is excellent that the reader can understand the importance of power transmission. The four phrases require references to support the opinions you have introduced.
Response 2: Thank you for the comment. We have included more references when describing the transmission of power.
Comment 3: In conclusion, you can present some ideas regarding future work.
Response 3: Thank you for the comment. We have added the future work.
Reviewer 2 Report
Comments and Suggestions for Authors
The article discusses an automatic method for span segmentation of powerline point cloud that accounts for adjacent powerline interference, aiming to provide "clean" data for the automatic reconstruction of powerline catenary curve model of each span. The structure of the manuscript is good. Abstract is correct. I suggest reducing the keywords. Keywords should be more concise.
Below are my comments on the manuscript:
1. The existing problems and research gaps must be clarified more explicitly. What are the new con-tributions to the scientific community? Moreover, the authors should briefly highlight their own achievements, findings, and the main contributions of the manuscript both at the end of the introduction and in the conclusions section.
2. I suggest adding a graphical abstract of the research methodology to the paper.
3. The bibliography includes only a few recently published papers (19 in total). Please consider adding more recent references.
4. The value of the article could be significantly enhanced if the authors expanded the discussion and included comparisons with results from other studies. Additionally, the results should be analyzed in greater detail, as the current analysis is too general.
5. In the conclusions, the authors should outline their plans for future research.
6. The presented results are typical of an engineering analysis and pertain only to the specific case analyzed.
7. Although there are some comments and minor shortcomings, the article discusses an interesting topic and could be considered for publication after revision.
Author Response
Comment 1: The existing problems and research gaps must be clarified more explicitly. What are the new con-tributions to the scientific community? Moreover, the authors should briefly highlight their own achievements, findings, and the main contributions of the manuscript both at the end of the introduction and in the conclusions section.
Response 1: Thank you for the comment. We have provided a clearer explanation of the existing problems and contributions. You can find the revision in the last two paragraphs of the introduction.
Comment 2: I suggest adding a graphical abstract of the research methodology to the paper.
Response 2: Thank you for the comment. To clarify the expression, we have included a diagram of the method.
Comment 3: The bibliography includes only a few recently published papers (19 in total). Please consider adding more recent references.
Response 3: Thank you for the comment. We have increased the number of recent publications.
Comment 4: The value of the article could be significantly enhanced if the authors expanded the discussion and included comparisons with results from other studies. Additionally, the results should be analyzed in greater detail, as the current analysis is too general.
Response 4: Thank you for the comment. We have elaborated on the discussion section. In addition, our work concentrates on the powerline lidar span segmentation after the point cloud classification with enormous noise. However, other works always treat the detected pylons as the ends of the power line directly. None of the papers open-sourced their code, which makes it difficult to reproduce on our actual data. To better demonstrate the robustness of our method, we have tested our method using more data, which you can find in Section 3.
Comment 5: In the conclusions, the authors should outline their plans for future research.
Response 5: Thank you for the comment. We have added one section to summarize the future work.
Comment 6: The presented results are typical of an engineering analysis and pertain only to the specific case analyzed.
Response 6: Thank you for the comment. We have conducted experiments using more data to verify the effectiveness of the algorithm.
Reviewer 3 Report
Comments and Suggestions for Authors
Dear Authors,
I have reviewed your manuscript and acknowledge the potential of the proposed solution for clustering LIDAR point clouds. In my opinion, this paper is suitable for publication, but before that happens, it requires addressing several key issues.
The submitted article represents an original contribution to science, particularly through the attempt to present a newly developed algorithm based on nearest neighbor search methods, which aims to optimize the processing of laser scanning data. Such solutions contribute to the advancement of optimization methods that leverage computational resources, ultimately shortening the time required for spatial data processing. These solutions are therefore desirable and expected both in academia and in the commercial market.
Regarding the cited literature, it is evident that the authors focused primarily on citing national experts, whereas optimization methods for LIDAR data processing are a core area of research worldwide. Given the international nature of the journal, this approach significantly narrows the perspective and should be expanded to include a solid review of global literature. I also believe that 19 references do not fully exhaust the topic and fail to provide sufficient motivation for undertaking this research. I request that you supplement the literature review and contextualize your presented research within a broader academic landscape.
Power line detection is one of the primary applications of laser scanning, whether terrestrial or aerial. Your manuscript lacks a broader market perspective. Many providers of geoinformation technology and software offer similar functionalities. Please compare your solution with available commercial software packages.
In geodetic measurements using laser scanning, the instrumentation plays a crucial role. The text lacks information regarding the selection of the data acquisition technology in the field. What specific measurement devices were used for your tests? What are their technical specifications, such as accuracy, resolution, scanner type (pulsed or phase-based), not to mention the specific model and manufacturer of the instrument? These are fundamental issues that should be the starting point for any methodological discussion.
The mathematical formulas presented in lines 148-151 should be accompanied by information regarding their source. Are they original equations developed by the authors or derived from a specific reference?
Subsections 3.1.1 and 3.1.2 contain only figures without any accompanying commentary. Each figure must be appropriately described and referenced within the text. Additionally, please clarify the significance of the various colors used in the figures. Do they convey specific information, or are they simply a result of random visualization settings?
Subsection 3.2.1, line 327, contains the enigmatic statement "Local Analysis" and, similarly, is followed by uncommented figures. This does not align with the standards of an international scientific journal and requires immediate correction.
Lines 346-351 in the "Conclusions" section contain information that should be placed in the abstract or introduction. After reading 15 pages of text, the reader is already well aware of the subject matter. The conclusion, however, fails to highlight the practical application of the developed method and its potential implementation in various software solutions.
In summary, the submitted manuscript requires a thorough restructuring and should be resubmitted for a second review. I wish the authors productive work!
Author Response
Comment 1: Regarding the cited literature, it is evident that the authors focused primarily on citing national experts, whereas optimization methods for LIDAR data processing are a core area of research worldwide. Given the international nature of the journal, this approach significantly narrows the perspective and should be expanded to include a solid review of global literature. I also believe that 19 references do not fully exhaust the topic and fail to provide sufficient motivation for undertaking this research. I request that you supplement the literature review and contextualize your presented research within a broader academic landscape.
Response 1: Thank you for the comment. Our approach focuses on span segmenting of the power lines after classification, in the presence of a large amount of noise, rather than designing an entire set of original power line point cloud processing system. We have included the relevant work in the introduction.
Comment 2: Power line detection is one of the primary applications of laser scanning, whether terrestrial or aerial. Your manuscript lacks a broader market perspective. Many providers of geoinformation technology and software offer similar functionalities. Please compare your solution with available commercial software packages.
Response 2: Thank you for the comment. We attempted to look for commercial software with the feature of span segmentation, but unfortunately, we couldn't find any. We also conducted tests on more data to verify our method.
Comment 3: In geodetic measurements using laser scanning, the instrumentation plays a crucial role. The text lacks information regarding the selection of the data acquisition technology in the field. What specific measurement devices were used for your tests? What are their technical specifications, such as accuracy, resolution, scanner type (pulsed or phase-based), not to mention the specific model and manufacturer of the instrument? These are fundamental issues that should be the starting point for any methodological discussion.
Response 3: Thank you for the comment. We have added the hardware information in Section 2.1.1.
Comment 4: The mathematical formulas presented in lines 148-151 should be accompanied by information regarding their source. Are they original equations developed by the authors or derived from a specific reference?
Response 4: Thank you for the comment. These formulas are designed to explain our own PCGrid method.
Comment 5: Subsections 3.1.1 and 3.1.2 contain only figures without any accompanying commentary. Each figure must be appropriately described and referenced within the text. Additionally, please clarify the significance of the various colors used in the figures. Do they convey specific information, or are they simply a result of random visualization settings?
Response 5: Thank you for the comment. We have revised the explanation in Sections 3.1.1 and 3.1.2.
Comment 6: Subsection 3.2.1, line 327, contains the enigmatic statement "Local Analysis" and, similarly, is followed by uncommented figures. This does not align with the standards of an international scientific journal and requires immediate correction.
Response 6: Thank you for the comment. We have corrected the writing.
Comment 7: Lines 346-351 in the "Conclusions" section contain information that should be placed in the abstract or introduction. After reading 15 pages of text, the reader is already well aware of the subject matter. The conclusion, however, fails to highlight the practical application of the developed method and its potential implementation in various software solutions.
Response 7: Thank you for the comment. We have included a description of the engineering significance in the conclusion.
Reviewer 4 Report
Comments and Suggestions for Authors
The article presents novel method for automatic span segmentation of powerline point clouds obtained from airborne LiDAR. The main challenge addressed is interference caused by adjacent or crossing powerlines during pylon and conductor extraction. To overcome this, authors introduced Point Count Grid (PCGrid) to accelerate DBSCAN clustering and three-step process. Important contribution of this work is the explicit handling of adjacent line interference—an aspect insufficiently explored in existing literature. Proposed framework enables a fully automated workflow, eliminating the need for manual preprocessing such as cropping, denoising or the incorporation of prior domain knowledge. Approach is validated on two real-world datasets (220 kV and 110 kV corridors)/
Despite of achievements there are some limitation of describe method because it was tested only on two datasets from a single region (Yunnan Province, China) without generalization from diverse environments. Additionally approach relies on a deep learning–based pre-classification of pylons and powerlines, which achieves an accuracy exceeding 95%. The study also lacks quantitative evaluation metrics beyond basic indicators such as pylon count and processing time. It should be underlined that highly noisy data, dense vegetation, or partial occlusions—are not explicitly analyzed.
It is recommended to improve manuscript consider the following issues:
1.Including segmentation accuracy against ground-truth spans instead of only reporting correct number of pylons.
2.Extending experiments to larger datasets with varied environments
3.Providing direct runtime and accuracy comparison with at least one state-of-the-art span segmentation approach.
4.Test sensitivity to classification errors using 5–10% misclassification of pylon points.
5,Considering integration of deep learning–based span detection as extension reducing reliance
There is also lack of discussion or consideration of alternative, simplified approaches based on LiDAR data that address environmental challenges in object recognition. It is recommended to include, for the benefit of readers, at least some methodologies demonstrating that 3D object detection can be effectively achieved using deep neural networks and Bird’s Eye View (BEV) representation of LiDAR point clouds (https://doi.org/10.5220/0010688400003063). Additionally, it would be good to mention possibility of integrating semantic (image-derived) and geometric (LiDAR-derived) information to improve span segmentation (https://doi.org/10.1007/978-3-031-71397-2_18) which can be considered for analysis.
From editorial point of view Figures 8–16 illustrate interference removal and segmentation steps but captions could better highlight achived improvements like reduction of false pylons.
Article can be published after addressing above improvements.
Author Response
Comment 1: Including segmentation accuracy against ground-truth spans instead of only reporting correct number of pylons.
Response 1: Thank you for the comment. Because the main-line powerlines and the main-line pylons are coupled. If the main-line point cloud is known, the span can be inferred based on the position of the power towers. If the position of the power pylons along the main line is known, the main-line point cloud and the actual spans can be obtained through the connectivity of the powerlines. Since the main-line point cloud in the power corridor is clearly visible, obtaining the sequence of the pylons along the main line will enable the correct span to be determined.
Comment 2: Extending experiments to larger datasets with varied environments.
Response 2: Thank you for the comment. We have conducted supplementary experiments on the 35kV data. Our approach is primarily designed for power corridor point clouds. This article also focuses mainly on this aspect. In the future, we can conduct analyses on point clouds in more scenarios (such as cities, roads).
Comment 3: Providing direct runtime and accuracy comparison with at least one state-of-the-art span segmentation approach.
Response 3: Thank you for the comment. Since the span segmentation is only one step in point cloud processing, and the current mainstream works mainly focus on power line reconstruction and do not have open-source codes, we directly compare the most commonly used point cloud indexing and clustering methods at present, ie. Kd-tree with DBSCAN.
Comment 4: Test sensitivity to classification errors using 5–10% misclassification of pylon points.
Response 4: Thank you for the comment. We have added Section 3.5 to test the robustness of our method.
Comment 5: Considering integration of deep learning–based span detection as extension reducing reliance
Response 5: Thank you for the comment. Certainly, the data-driven approach is currently the mainstream direction. We are considering incorporating the learning-based methods into our future research plans.
Comment 6: There is also lack of discussion or consideration of alternative, simplified approaches based on LiDAR data that address environmental challenges in object recognition. It is recommended to include, for the benefit of readers, at least some methodologies demonstrating that 3D object detection can be effectively achieved using deep neural networks and Bird’s Eye View (BEV) representation of LiDAR point clouds (https://doi.org/10.5220/0010688400003063). Additionally, it would be good to mention possibility of integrating semantic (image-derived) and geometric (LiDAR-derived) information to improve span segmentation (https://doi.org/10.1007/978-3-031-71397-2_18) which can be considered for analysis.
Response 6: Thank you for the comment. Our approach mainly focuses on the extraction and span segmentation of the main line of the power corridor point clouds, and has a relatively weak connection with point cloud object detection.
Comment 7: From editorial point of view Figures 8–16 illustrate interference removal and segmentation steps but captions could better highlight achived improvements like reduction of false pylons.
Response 7: Thank you for the comment. We have revised the explanations for these diagrams in Section 3.
Round 2
Reviewer 3 Report
Comments and Suggestions for Authors
Dear authors,
Thank you for submitting a revised version of the manuscript. Reading your text, one can see that you have put a lot of work into improving the previous version, which can be appreciated. I also consider the answers to my questions and suggestions to be sufficient, although I will not agree with the fact that similar studies are difficult to find. On the contrary, there are a lot of them – in fact, it is worth asking any manufacturer of UAV-based technology in order to get a comprehensive answer. I am not mentioning that such research is conducted at many universities around the world. Perhaps you haven't come across them because, as far as I can see, you rely mainly on local authors! I propose a broader perspective – next time, I would do the literature review based on the published outcomes from both Europe and the USA. I don't want to advertise my parent scientific unit when doing this review, but please believe me – we have been doing very similar works for many years. Nevertheless, I believe that your text is worth publishing, at least because of its significant contribution to enriching the current state of knowledge in this area.
Thank you for taking my comments into account and making the necessary corrections to your text. I wish you good luck and further fruitful work!
Author Response
Comment 1: Thank you for submitting a revised version of the manuscript. Reading your text, one can see that you have put a lot of work into improving the previous version, which can be appreciated. I also consider the answers to my questions and suggestions to be sufficient, although I will not agree with the fact that similar studies are difficult to find. On the contrary, there are a lot of them – in fact, it is worth asking any manufacturer of UAV-based technology in order to get a comprehensive answer. I am not mentioning that such research is conducted at many universities around the world. Perhaps you haven't come across them because, as far as I can see, you rely mainly on local authors! I propose a broader perspective – next time, I would do the literature review based on the published outcomes from both Europe and the USA. I don't want to advertise my parent scientific unit when doing this review, but please believe me – we have been doing very similar works for many years. Nevertheless, I believe that your text is worth publishing, at least because of its significant contribution to enriching the current state of knowledge in this area.
Response 1: Thanks for the comment. Indeed, we mostly use local software in our daily data processing, which results in a slower pace of keeping up with the latest international research developments. High-precision point cloud classification has been integrated into many software applications, but our work focuses on span segmentation with noise and can be applied to coarsely labeled data. We fully agree with your suggestion. Considering the timeliness of responding to the reviewers' comments and the copyright issues related to using the new business software, we incorporated a wider range of data to demonstrate the robustness of our method.
Comment 2: I encourage the author to add some international positions to the reference list. It will certainly strengthen the article.
Response 2: Thanks for the comment. We have added some international research to the reference list.